# Multiple Myeloma in Patients over 80: A Real World Retrospective Study of First Line Conservative Approach with Bortezomib Dexamethasone Doublet Therapy and Mini-Review of Literature

**DOI:** 10.3390/cancers14194741

**Published:** 2022-09-28

**Authors:** Laurence Huynh, Rudy Birsen, Lucie Mora, Anne-Laure Couderc, Nathalie Mitha, Anaïs Farcet, Amale Chebib, Pascal Chaibi

**Affiliations:** 1Service d’Hématologie et Oncologie Gériatrique, Hôpital Charles Foix, APHP, 94200 Ivry Sur Seine, France; 2Service d’Hématologie, Hôpital Cochin, APHP, Université de Paris Cité, 75014 Paris, France

**Keywords:** multiple myeloma, elderly, FRAIL

## Abstract

**Simple Summary:**

Multiple Myeloma (MM) is frequent and represents 2% of all cancers. Daratumumab bortezomib-melphalan-prednisone (D-VMP) and daratumumab lenalidomide dexamethasone (D-Rd) are considered the standard of care for elderly patients with newly diagnosed MM (NDMM), defined as transplant-ineligible patients over 65 years. However, the “elderly” patient population is heterogeneous, and prospective trials exclude the oldest and frailest patients because of co-morbidities or altered Eastern Cooperative Oncology Group Performance Status (ECOG PS). According to the IMWG frailty score, patients over 80 are considered as frail. Few data are available on octogenarian patients with NDMM, and their optimal management remains controversial. We here report one of the largest retrospective series investigating doublet therapy with bortezomib dexamethasone (Vd) as the first-line treatment for unselected octogenarian patients with NDMM.

**Abstract:**

Data on octogenarian patients with MM are scarce, and optimal management remains controversial. We report a retrospective cohort of unselected octogenarian patients with NDMM treated with bortezomib dexamethasone (Vd). Seventy-four patients were treated with an initial doublet therapy (Vd regimen, 2–3 cycles, induction). A dose escalation with an adjunction of melphalan or cyclophosphamide was proposed for patients who had an insufficient response after induction and who could tolerate it. In responders, the treatment was continued until progression or a plateau response for 6 months (consolidation). The overall response rate was 73%. After a median follow-up of 31.4 months, median progression-free survival (PFS) and overall survival (OS) were 13.2 and 26.9 months, respectively. PFS and OS of patients with ECOG PS < 3 (25.4 and 54.9 months, respectively) were better in comparison to PFS and OS of patients with ECOG PS ≥ 3 (9.3 and 11.3 months, respectively). Thirteen patients (17.6%) died during induction. Twelve patients (16.2%) died during consolidation. In conclusion, a conservative therapeutic strategy based on Vd resulted in a good response rate. However, the survival remains poor in the population of patients with an ECOG PS ≥ 3, mainly because of early mortality not related to progressive disease.

## 1. Introduction

Multiple myeloma (MM) is a hematological malignancy characterized by an abnormal clonal plasma cell infiltration in the bone marrow. Before 2019, bortezomib-melphalan-prednisone (VMP) and lenalidomide dexamethasone (Rd) were considered the standard of care for elderly patients with newly diagnosed multiple myeloma(NDMM), defined as transplant-ineligible patients over 65 years [1,2]. Three recent phase III trials have changed this standard of care for patients with NDMM ineligible or without an intent for immediate autologous stem cell transplant (ASCT). The SWOG S0777 (comparing bortezomib, lenalidomide, and dexamethasone (VRd) with Rd) [3,4], the ALCYONE (comparing Daratumumab VMP with VMP) [5,6], and the MAIA (comparing Daratumumab Rd with Rd) [7,8] studies showed an improvement in PFS and OS in the investigational arm compared to the control arm. Thus, the latest ESMO recommendations identify DaraRd, DaraVMP, or VRd as the frontline therapy options for NDMM not eligible for ASCT [9].

However, the “elderly” patient population is heterogeneous, and most of the prospective trials exclude the oldest and frailest patients because of comorbidities or altered Eastern Cooperative Oncology Group Performance Status (ECOG PS). Thus, data on super elderly patients (over 80) with NDMM are scarce, and optimal management remains controversial. This contrasts with the high incidence of MM in this population, which represents a third of NDMM in France [10]. We here report a large retrospective cohort investigating doublet therapy with bortezomib dexamethasone (Vd) as first-line treatment for unselected super elderly patients with NDMM.

## 2. Materials and Methods

### 2.1. Patients

We retrospectively reviewed all consecutive NDMM patients aged 80 years or older admitted to the geriatric oncology unit of Charles-Foix Hospital between April 2013 and October 2019. We excluded the few patients with a pre-existing severe neuropathy that contraindicated the use of bortezomib. The search for high-risk MM criteria was not performed in our cohort.

### 2.2. Treatment

Patients were treated with an initial doublet therapy: bortezomib subcutaneous 1.3 mg/m^2^ (SC) plus oral dexamethasone 20 mg (Vd regimen). Induction consisted of 2 to 3 cycles of Vd (Day 1–4–8–11; 21-day cycle) followed by maintenance with weekly Vd (28-day cycle). A dose escalation with VMd (adjunction of Melphalan 0.2 mg/kg day 1–4 every 6 weeks) or VCd regimen (adjunction of Cyclophosphamide 750 mg/m^2^ I.V. day 1, capped at 1000 mg total dose) was proposed for patients who had a stable disease (SD), minimal response (MR) or partial response (PR) after induction and who could tolerate it. If a response was observed, treatment was continued until progression or a plateau response for 6 months. Bortezomib doses were adjusted according to the toxicities observed. Antibiotics (trimethoprim-sulfamethoxazole and amoxicillin) and antiviral prophylaxis (acyclovir) were given during the whole treatment. Response to treatment was assessed using criteria based on the IMWG uniform response criteria [11].

### 2.3. Geriatric Assessment

At baseline a geriatric assessment was performed: Katz’s Activities of Daily Living (ADL), nutritional evaluation (body mass index (BMI) and albumin serum value), walking with assistance, ECOG PS (5-point scale, with higher numbers indicating greater disability), and age-adjusted Charlson comorbidity index (aaCCI).

### 2.4. Outcomes

Overall response rate (ORR) was defined as a PR or better. PFS was defined by the time from diagnosis to the date of progression or death. OS was defined by the time from diagnosis to death from any cause or until the last follow-up.

### 2.5. Statistical Analysis

Survival curves were estimated using the Kaplan–Meier method. Prognostic factor analyses were performed in univariate analysis with the Log Rank test and in multivariate analysis using a Cox model. *p* values < 0.05 were considered statistically significant.

## 3. Results

From April 2013 and October 2019, a total of 74 patients were treated. Patient’s characteristics are listed in Table 1.

The median age was 85 (range, 80–95 years), and 18% of patients were over 90. As expected, patients presented several comorbidities: arterial hypertension (66%), heart disease from any cause (45%), neurological vascular or neurodegenerative disease (21.6%), renal failure (18%), diabetes (16%), and chronic pulmonary disease (9%). ECOG PS was ≥3 in 58% of patients and 46% of patients showed severe renal impairment (defined as Creatinine Clearance according to Cockcroft–Gault equation <30 mL/min). Fifty-six patients (76%) had an aaCCI > 5 and 35% were ADL dependent (i.e., ADL < 4.5). All patients were considered frail based on the IMWG frailty score [12] and the simplified frailty scale [13]. According to the simplified frailty scale, 3 patients (4%) had a score of 2, 11 patients (15%) a score of 3, 42 patients (57%) a score of 4, and 18 patients (24%) a score of 5.

### 3.1. Response

The best overall response rate (ORR) achieved during induction was 64.9%, including 14 (19%) very good partial response (VGPR) and 34 (45.9%) PR. The best ORR achieved during the first line of treatment was 73%, including 25 (33.8%) very good partial response (VGPR) and 29 (39.2%) PR (Table 2). Nineteen patients (25.7%) underwent dose escalation after induction. Among them, five (26.3%) patients were in PR, six (31.6%) in MR, and eight (42.1%) in SD. One (5.3%) achieved VGPR, ten (52.3%) PR, four (21.2%) MR, and four (21.2%) SD (Table 2).

### 3.2. PFS and OS

The median follow-up was 31.4 (4.1–59.9) months. The median PFS and OS were 13.2 and 26.9 months, respectively (Figure 1a,b). The two-year PFS and OS were 32.1% (CI 95%, 20.8–44) and 51.6% (CI 95%, 39.5–62.4), respectively. ECOG PS ≥ 3 was significantly associated with a lower PFS (median 9.3 vs. 25.4 months, *p* < 0.0027) and estimated OS (median 11.3 vs. 54.9 months, *p* = 0.0002) (Figure 1c,d).

Several factors were associated with inferior survival in univariate analysis including ECOG PS ≥ 3 (*p* = 0.0002), ADL < 4.5 (*p* = 0.0002), BMI < 21 (*p* = 0.028), and walking with assistance (*p* = 0.0033) (Table 3). Only ECOG PS ≥ 3 was still significant in multivariate analysis (*p* < 0.01).

### 3.3. Tolerability and Safety

Patients received a median of 9 (1–25) cycles of bortezomib and 53 patients underwent dose adjustment, mostly due to recurrent grade 1–2 thrombocytopenia (Table 4).

Ten patients (13.5%) discontinued treatment for major toxicity: nine patients (12.2%) for grade 3–4 neurological toxicity and one patient for grade 3 digestive toxicity. Thirteen patients (17.6%) died during induction. The causes of death were severe acute respiratory failure (pulmonary embolism/infection/acute cardiovascular failure) (*n* = 6), hemorrhagic shock (*n* = 1), subdural hematoma (*n* = 1), unknown causes (*n* = 1), and infection (*n* = 4). Twelve patients (16.2%) died during consolidation. The causes of death were hemorrhagic shock (*n* = 1), acute cardiovascular failure (*n* = 2), unknown cause (*n* = 4), and infection (*n* = 5). Most deaths occurred in patients with initial ECOG PS ≥ 3 (*n* = 21/25, 84%). Hematological toxicity was reported for most of the patients: 75.7% of grade 1–2 thrombocytopenia, 2.7% of grade 1–2 neutropenia, 6.7% of grade 1–2 anemia, and 6.7% of grade 3–4 anemia. We observed 17.6% of grade 3–4 infections, 2.7% of grade 3–4 cardiac toxicity, and 10.8% of grade 1–2.

## 4. Discussion

Our results in terms of PFS and OS are inferior compared to most of the prospective trials [1,2,3,5,7,14,15,16]. However, most of these studies excluded patients with major comorbidities, ECOG PS ≥ 3, and patients with impaired renal function. If we focus on the patients with PS < 3, our PFS (25.4 months) and OS (54.9 months) are similar to those found with the prospective evaluation of revlimid and dexamethasone (respectively, 25.5 and 58.9 months in the FIRST trial [2]). In less-selected patient cohorts, the results are similar to ours (median PFS and OS were 7–28.8 and 21–54.6 months, respectively) [17,18,19,20,21,22,23,24]. We summarized data from prospective and retrospective studies focusing on elderly patients with MM in Table 5.

Our choice to propose a bortezomib dexamethasone doublet therapy over lenalidomide dexamethasone was based on several factors related to the specificities of our super elderly patients. An important issue is the adherence to oral cancer therapy in older adults. Adherence may be compromised specifically in the presence of cognitive impairment, multimorbidity, and polypharmacy [27,28]. These relevant risk factors of poor adherence are particularly prevalent in our patients. Thus, the use of bortezomib instead of lenalidomide was a simple way to ensure adherence to treatments. In addition, nearly half of the patients had severe renal impairment. Although dose adjustment with lenalidomide was possible, we believed that the efficacy and toxicity profiles were uncertain in these patients, as was later demonstrated [29]. Finally, even if the administration of bortezomib was more restrictive than lenalidomide, it indirectly induced a closer medical follow-up of these frail patients.

Our conservative approach aimed to offer a treatment of moderate intensity (doublet therapy) during the first courses to improve ECOG PS of the patient for a dose escalation if necessary (triplet therapy). This assumption that “more treatments do not necessarily translate into better survival in super elderly/very frail patients with NDMM” was confirmed by two prospective trials that demonstrated that the addition of melphalan or endoxan to Vd did not provide benefit over Vd in frail patients. In the UPFRONT trial, the doublet regimen Vd showed lower toxicity and similar efficacy to the triplet regimen (with melphalan or thalidomide) [25]. Larocca and al., in a prospective trial designed to include very elderly and frail patients, showed that VMP led to treatment discontinuation for toxicity in 20% [14]. There was no substantial advantage for a three-drug regimen (VMP or VCP) compared to VP (median PFS 14 months, 2-year OS 60%), due to the higher toxicity and treatment discontinuation in the three-drug regimen.

However, this conservative approach seems outdated according to the current paradigm of survival improvement in patients with NDMM ineligible or without an intent for immediate autologous stem cell transplant (ASCT), which relies on the addition of new therapies to a pre-existing backbone. This is illustrated by the SWOG S0777, ALCYONE, and MAIA studies. The SWOG S0777, a randomized phase III trial, compared VRd with Rd [3,4]. PFS (median PFS 41 versus 29 months, *p* = 0.003) and OS (median OS not reached versus 69 months, *p* = 0.0114) were improved with the addition of bortezomib to Rd [4]. Notably, only 43% of the patients in this study were older than 65 years. A recent real-world analysis of patients who received VRd as their first line of therapy showed a median PFS of 26.5 months, which was markedly shorter than that observed in the SWOG S0777 trial [26]. Patients in this real-world analysis were older and a higher proportion was frail compared with the SWOG S0777 study. It is likely that the addition of lenalidomide to Vd is too toxic in a frail patient population to confer an advantage in survival.

The addition of daratumumab to VMP (D-VMP) and Rd was explored in the ALCYONE and MAIA trials. The ALCYONE trial showed that PFS was significantly longer with D-VMP (mPFS 36.4 months) versus VMP alone (mPFS 19.3 months). The Kaplan–Meier estimate of the 36-month rate of OS was 78% in the D-VMP group and 67.9% in the VMP group. The HR for death in the D-VMP group compared with the VMP group was 0.6 (*p* = 0.0003). The MAIA study showed that the risk of disease progression or death was significantly lower among those who received daratumumab plus Rd (D-Rd) than among those who received Rd alone [7]. At a median follow-up of 56.2 months, median PFS was not reached in the D-Rd group versus 34.4 months in the Rd group (HR 0.53; *p* < 0.0001). The Kaplan–Meier estimate of the 60-month rate of OS was 52% in the D-Rd group and 28.7% in the Rd group. Median OS was not reached in either group (HR 0.68; *p* = 0.0013) [7,8]. In prespecified subgroup analyses of these studies, the addition of daratumumab significantly improved PFS in patients older than 75 years [5,7]. A frailty subgroup analysis using the simplified frailty scale [13] was conducted for both studies. The PFS benefit of daratumumab addition was observed in all frailty subgroups, supporting the clinical benefit of daratumumab addition to VMP or Rd in NDMM patients regardless of frailty status. However, these results should be extrapolated with caution to the super elderly/very frail patients that were not represented in these trials. Thus, only 15 patients (20.3%) of our cohort could have been included in the MAIA trial considering the main exclusion criteria (cognitive disorders, ECOG PS ≥ 3, Creatinine clearance < 30 mL/min).

Anti-CD38 monoclonal antibodies were not available as first-line therapy during the period covered by our study. However, in our opinion, it remains to be demonstrated that its addition improves the prognosis of super elderly/very frail patients in combination with a proteasome inhibitor, considering that daratumumab may significantly increase some adverse events. For example, grade 3/4 pneumonia was almost two (13.7 versus 7.9% in MAIA) to three (11.3 versus 4% in ALCYONE) times more frequent with daratumumab [5,7]. This did not lead to the discontinuation of treatment in these trials, including in the group of patients considered to be frail. However, for super elderly/very frail patients, any increase in adverse events may have greater clinical consequences. The recently reported results of the Hovon 143 study illustrate this point [24]. This phase II trial was specifically designed for frail patients (median age 81) and evaluated the efficacy and tolerability of ixazomib-daratumumab-low-dose-dexamethasone. The ORR during induction was 78%, the median PFS was 13.8 months, and OS at 12 months was 78%. These disappointing results were explained by the high rate of induction therapy discontinuation (51% of patients; progression for 19%, toxicity for 9%, death for 9%, non compliance for 6% and other causes for 8%) that negatively influenced PFS and OS.

This highlights one of the limitations in defining frailty through geriatric assessment in a real-world setting. Geriatric assessment and in particular the definition of the frailty profile is a recent concern of the collaborating groups behind the major therapeutic trials in MM. Based on a pooled analysis of 869 individual newly diagnosed elderly patient data from three prospective trials, the IMWG proposes a score for the measurement of frailty that combines age, functional status (using Katz Activity of Daily Living and the Lawton Instrumental Activity of Daily Living) and comorbidities [12]. One of the main limitations of the IMWG frailty score is the fact that functional status assessment is a time and manpower-consuming procedure. To overcome this limitation, a simplified frailty scale (using scores for age, CCI, and ECOG PS) was built on the cohort of transplant-ineligible patients with NDMM treated in the FIRST trial. Both of these frailty scales can predict survival and toxicity in patients with transplant-ineligible NDMM. Notably, these scores were based on cohorts of patients from prospective phase III trials that excluded even more frail patients from the outset: patients with significant comorbidities, renal failure, and/or ECOG PS ≥ 3. These patients, overrepresented in our study, experienced a high rate of early death, mostly due to infection or multifactorial respiratory failure (cardiovascular failure, infection, or pulmonary embolism).

Other ways than simply adding the latest therapy should be explored to improve management of this population. Considering the toxicity profile of the drugs we used, it is likely that dexamethasone is one of the main causes of the observed morbidity and mortality. A previous study has shown that morbidity was significantly higher with a dexamethasone-based regimen compared to a prednisone-based regimen [30]. Thus, the use of prednisone rather than low-dose dexamethasone may limit treatment-related morbidity and mortality and may increase the tolerance of therapy in our population. Dexamethasone-free regimens, such as those currently investigated by the Intergroupe Francophone du Myelome-study group (ClinicalTrials.gov identifier: NCT03993912), are also promising strategies.

Another approach to reduce mortality could be to improve the antimicrobial prophylaxis during treatment, considering that infection contributed to a third of early mortality in our cohort. A phase III study reported that the addition of prophylactic levofloxacin to active MM treatment during the first 12 weeks of therapy significantly reduced febrile episodes and deaths compared with placebo without increasing healthcare-associated infections [31]. However, patients older than 75 years were excluded and these results may not be replicated in an older population for whom the side effects of fluoroquinolones may be increased by associated comorbidities [32].

Finally, given the poor results of patients with an ECOG PS ≥ 3, some might argue that treating such frail and elderly patients is not relevant nor ethical. However, one-third of these patients experienced long-term OS and all patients had a symptomatic MM that could result in an impaired quality of life and short-term death if left untreated. This suggests that all elderly patients should be treated, considering that a patient with ECOG PS ≥ 3 initially may have long-term survival and that we are not able to reliably predict this.

## 5. Conclusions

We report one of the largest retrospective cohorts of super elderly/very frail patients with NDMM homogeneously treated with a first-line bortezomib dexamethasone doublet therapy. Our conservative therapeutic strategy resulted in an overall response rate of 73%. After a median follow-up of 31.4 months, median PFS and OS were 13.2 and 26.9 months, respectively. OS at 12 months was 62.2%. The purpose of our study is not to assert that our therapeutic strategy is the reference approach for super elderly/very frail patients. The PFS and OS are certainly insufficient, in particular for the 20% of patients for whom the results of the MAIA trial can be applied. However, these results constitute a basis for the evaluation of new therapeutic strategies, in particular those involving monoclonal antibodies (i.e., anti-CD38 monoclonal antibodies), and invite reflection on ways to improve management in a super elderly/very frail patient population that is mostly excluded from prospective trials.

## Figures and Tables

**Figure 1 cancers-14-04741-f001:**
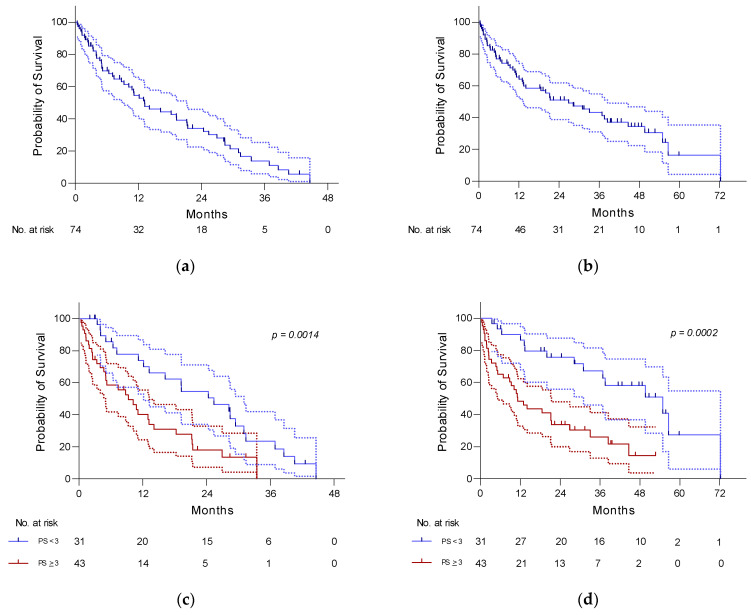
Progression-free survival (**a**) and overall survival (**b**) of the whole cohort. Progression-free survival (**c**) and Overall survival (**d**) in the group of patients with ECOG PS < 3 (blue line) and ≥3 (red line).

**Table 1 cancers-14-04741-t001:** Patients’ characteristics and geriatric assessment of the study cohort.

Baseline Characteristics	*n* = 74	%
Median age (years)	85 (80–95)	-
Woman	40	54
Myeloma type		
IgG	48	65
IgA	10	13
Light chains	16	22
Hemoglobin < 10 g/dL	64	86
Calcium > 2.75 mmol/L	19	26
Clearance Cockcroft < 30 mL/min	34	46
GFR CKD EPI < 30 mL/min/1.73 m^2^	25	34
Bone damage	40	54
Performance Status		
<3	31	42
≥3	43	58
Arterial hypertension	49	66
Chronic Heart Failure	6	8
Ischemic heart disease	8	11
Atrial fibrillation	14	19
Valvular heart disease	5	7
Diabetes	12	16
Chronic Kidney Failure	13	18
Chronic respiratory deficiencies	7	9
Cognitive disorders	11	15
History of Stroke	5	7
Albumin < 35 g/L	41	55
ADL		
<4.5	23	31
≥4.5	51	69
Walking with assistance		
Without	45	61
With	29	39
Body Mass Index		
<21 kg/m^2^	19	25.7
≥21 kg/m^2^	55	74.3
Age-adjusted Charlson Comorbidity Index		
<5	18	24
≥5	56	76

**Table 2 cancers-14-04741-t002:** Response rate of the study cohort.

Response Assessment	*n* = 74	%
Therapeutic responses after induction		
VGPR	14	19
PR	34	45.9
MR	13	17.6
SD	3	4
PD	1	1.3
Not evaluated	9	12.2
Overall response rate (CR + VGPR + PR)	64.9%	
Best response during treatment		
VGPR	25	33.8
PR	29	39.2
MR	4	5.4
SD	6	8.1
PD	1	1.3
Not evaluated	9	12.2
Overall response rate (CR + VGRP + PR)		73%

**Table 3 cancers-14-04741-t003:** Results of univariate statistical analysis. *: *p* < 0.05.

Prognostic Factors	OS		PFS	
	HR (95% CI)	*p* Value	HR (95% CI)	*p* Value
Sex (female vs. male)	0.94 (0.53 to 1.67)	0.82	0722 (0.40 to 1.28)	0.23
Age-Adjusted CCI (>5 vs. < 5)	1.12 (0.58 to 2.15)	0.74	1.04 (0.53 to 2.03)	0.91
ECOG PS (≥3 vs. <2)	2.88 (1.61 to 5.15)	0.0002 *	2.13 (1.24 to 3,66)	0.0027 *
Heart disease	1.38 (0.76 to 2.51)	0.26	1.54 (0.84 to 2.84)	0.14
Neurological disease	0.70 (0.36 to 1.40)	0.34	0.76 (0.38 to 1.53)	0.48
Lung disease	1.43 (0.54 to 3.83)	0.41	1.00 (0.43 to 2.36)	0.99
Comorbidities > 3 vs. <3	1.16 (0.65 to 2.05)	0.62	1.08 (0.61 to 1.92)	0.79
Albumin < 35 g/L vs. >35 g/L	1.15 (0.65 to 2.04)	0.62	1.32 (0.45 to 2.33)	0.34
Hemoglobin < 10 g/dL vs. >10 g/dL	1.33 (0.61 to 2.89)	0.51	1.08 (0.52 to 2.24)	0.83
Calcium > 2.75 mmol/L vs. <2.75 mmol/L	0.71 (0.37 to 1.34)	0.33	0.77 (0.40 to 1.47)	0.46
Creatinine level > 177 µmol/L vs. <177 µmol/L	1.42 (0.72 to 2.80)	0.23	1.60 (0.79 to 3.24)	0.13
Clearance Cockcroft < 30 mL/min vs. >30 mL/min	1.20 (0.67 to 2.13)	0.53	0.78 (0.46 to 1.33)	0.86
GFR CKD < 30 mL/min vs. >30 mL/min	1.54 (0.81 to 2.94)	0.14	1.46 (0.79 to 2.69)	0.16
ADL < 4.5 vs. >4.5	3.16 (1.26 to 7.93)	0.0002 *	2.0 (1.074 to 3.72)	0.0328 *
BMI < 21 vs. ≥21	1.89 (0.95 to 3.73)	0.028 *	1.28 (0.70 to 2.33)	0.39
Walking without assistance vs. with assistance	2.01 (0.87 to 4.67)	0.0033 *	1.61 (0.92 to 2.84)	0.073

**Table 4 cancers-14-04741-t004:** Treatment dose adaptations.

	*n* = 74
Median number of bortezomib cycle	9
Total number of bortezomib cycle	-
<5	21
5–10	28
>10 (maximum 25)	25
Bortezomib dose adaptation	*n* = 53 (71.6%)
1 mg/m^2^	53 (71.6%)
0.7 mg/m^2^	34 (46%)
Dexamethasone interruption	*n* = 10 (14.3%)
Temporary	1 (1.3%)
Permanent	9 (12%)

**Table 5 cancers-14-04741-t005:** Summary of data from prospective and retrospective studies focusing on elderly patients with NDMM. NR: Not Reported; Vd: Bortezomib + low dose dexamethasone; VMP: Bortezomib Melphalan Prednisone; MP: Melphalan Prednisone; Rd: Lenalidomide dexamethasone; MPT: Melphalan prednisone Thalidomide; DVMP: Daratumumab VMP; DRd: Daratumumab Rd; D-Ix-d: Daratumumab Ixazomib Dexamethasone.

	Study	No. of Patients	Median Age (Years)	PS ≥ 3	Treatment	ORR (%)	PFS (Median, Months)	OS (Median, Months)
**Prospective**	Upfront [25]	502	73	0	Vd/VTD/VMP	73/80/70	14.7/15.4/17.3	49.8/51.5/53.1
	Vista [1]	682	71	0	VMP/MP	71/35	24/11.6	NA/43
	First [2]	1623	73	0.006	Rd/MPT	75/62	25.5/21.2	59.1/49.1
	Larocca [14]	152	78	0	VP/VCP/VMP	64/67/86	14/15.2/17.1	60/70/76% at 24 months
	O’Donnel [15]	50	73	0	VRD lite	86	35.1	Not reached
	SWOG [3,4]	525	63	0	VRd/Rd	82/72	41/29	Not Reached/69
	ALCYONE [5,6]	706	71	0	DVMP/VMP	90.9/73.9	36.4/19.3	Not reached
	MAIA [7]	737	74	0	DRd/Rd	92.9/81.3	Not Reached/31.9	70.6/55.6% at 30 months
	Larocca [16]	199	75	0	Rd/Rd-R	78/68	20.2/18.3	74/63% at 36 months
	Stege [24]	65	81	8	DIxd	78	13.8	62% at 22.9 months
**Retrospective**	Our study	74	85	58	Vd	73	13.2	26.9
	Matsue [19]	42	85	52.4	Bortezomib +ImiD based	88.1	19.1	31.9
	Ediriwickrama [23]	52	79	25	T/Vd/Rd	70	NR	36
	Chan [20]	155	76	NR	VCD	79.4	21.7	45.1
	Gavriatopoulo [22]	110	83	66	VR/V/R	63	7	21
	Panistas [21]	89	87	NR	Bortezomib +ImiD based	NR	11.7	22.2
	Dimopoulos [22]	155	82	<60	Bortezomib +ImiD based	58	NR	22
	Bang [18]	139	80	<32	Bortezomib +ImiD based	51	20	27
	Medhekar [26]	2342	67	>3.4	VRd	NR	26.5	NR

## Data Availability

The data can be shared up on request.

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
