# Peer review of "Multiple Myeloma in Patients over 80: A Real World Retrospective Study of First Line Conservative Approach with Bortezomib Dexamethasone Doublet Therapy and Mini-Review of Literature"

_cancers, 2022, doi:10.3390/cancers14194741_

Round 1
Reviewer 1 Report
Huynh, L, et al. reported the treatment outcome of octogenarian patients with multiple myeloma (MM). The authors retrospectively analyzed the background and prognosis in 74 patients treated with bortezomib and dexamethasone (Bd) therapy. They found that the overall response rate was 73%, and the median progression-free survival and overall survival was 13.2 and 26.9 months, respectively. In terms of prognostic factors, performance status (≥3) was a significant independent factor for overall survival. As for safety, thrombocytopenia mostly affected dose adjustment of Bd therapy. Most deaths occurred in patients with PS ≥3 and not related to progression of MM. Based on these data, they concluded that all elderly patients with MM should be treated because of prolonged survival in some patients even PS ≥3.
1. Additional discussion on indication of Bd and Rd therapy as 2-drug therapy for elderly patients would be appreciated in terms of efficacy and safety.
2. It is recommended to discuss the usefulness of Geriatric assessment for prediction of prognosis.
3 . In Table 1, n=74 and % of Geriatric assessment are not required.
4. In Table 2, VGRP is a typographical error. What is NED?
5. In Discussion, table 5 is a typographical error.
6. Trade names such as Velcade and Revlimid should be unified into bortezomib and lenalidomide throughout the manuscript.
Author Response
Huynh, L, et al. reported the treatment outcome of octogenarian patients with multiple myeloma (MM). The authors retrospectively analyzed the background and prognosis in 74 patients treated with bortezomib and dexamethasone (Bd) therapy. They found that the overall response rate was 73%, and the median progression-free survival and overall survival was 13.2 and 26.9 months, respectively. In terms of prognostic factors, performance status (≥3) was a significant independent factor for overall survival. As for safety, thrombocytopenia mostly affected dose adjustment of Bd therapy. Most deaths occurred in patients with PS ≥3 and not related to progression of MM. Based on these data, they concluded that all elderly patients with MM should be treated because of prolonged survival in some patients even PS ≥3.
We thank reviewer 1 for his positive comments and feedback that helped us improve the manuscript. We hope that the changes made will meet his expectations.
- Additional discussion on indication of Bd and Rd therapy as 2-drug therapy for elderly patients would be appreciated in terms of efficacy and safety.
As requested by reviewer 1, we have added paragraphs to the discussion about our motivations for offering Vd to our patients rather than Rd. We have also added a discussion on the efficacy and safety of adding Daratumumab in frail elderly patients.
- It is recommended to discuss the usefulness of Geriatric assessment for prediction of prognosis.
A discussion of the value and limitations of geriatric assessment was added to the discussion.
3 . In Table 1, n=74 and % of Geriatric assessment are not required.
We thank reviewer 1 for noting this error which has been changed.
- In Table 2, VGRP is a typographical error. What is NED?
We thank reviewer 1 for noting this error which has been changed.
- In Discussion, table 5 is a typographical error.
We thank reviewer 1 for noting this error which has been changed.
- Trade names such as Velcade and Revlimid should be unified into bortezomib and lenalidomide throughout the manuscript.
We thank reviewer 1 for noting this error which has been changed.

Reviewer 2 Report
The Authors retrospectively analysed a cohort of 74 newly diagnosed elderly MM patients treated with doublet Vd between 2013 and 2019 in a geriatric oncology unit. The study is not original and seems to be outdated since treatment of newly diagnosed MM has changed a lot in recent years.
- In the introduction the Authors state that VMP and Rd are considered as the standard of care for newly diagnosed patients who are ineligible for ASCT but this statement is no longer true since DRd, D-VMP and VRd are the new standard therapies as suggested by the most recent ESMO guidelines (Dimopoulos et al, Ann Oncol 2021)
- It is true that management of octogenarian patients can be difficult due to comorbidities or altered performance status but we know that in frail patients, defined according to simplified frailty scale, who received DRd median PFS was NR vs 30.4 months in patients receiving Rd (Facon et al, Leukemia 2021, not mentioned in the paper). Safety profile was acceptable and infections leading to discontinuation were rare and pneumonia was a reason of discontinuation only in 2 patients in the frail group (1.2%) receiving DRd.
- How many patients received Rd regimen in the same period and, in addition to peripheral neuropathy, what were other criteria for choosing Vd or Rd considering that this latter is an oral effective regimen, well tolerated and very easy to administer in an unfit population?
- I agree that all elderly patients should be treated (see the last statement of discussion section) but I think that these patients should receive the most effective regimen since if they relapse probably will not be able to receive further therapies
Author Response
The Authors retrospectively analysed a cohort of 74 newly diagnosed elderly MM patients treated with doublet Vd between 2013 and 2019 in a geriatric oncology unit.
We thank reviewer 2 for his comments and feedback that helped us improve the manuscript. based on his comments, we have largely modified the discussion. We hope that the changes made will meet his expectations.
The study is not original and seems to be outdated since treatment of newly diagnosed MM has changed a lot in recent years.
We agree with reviewer 2 that this study is not the most attractive since it does not present any revolutionary therapeutic strategy. However, as we have tried to justify in the new discussion that we propose, we believe that our data are still of interest.
Indeed, it presents a homogeneous cohort of super elderly/very frail patients who are excluded from prospective trials, for which data are sparse. The results of MAIA and ALCYONE are indeed very promising for frail patients. However, these results should be extrapolated with caution to the super elderly/very frail patients that were not represented in these trials. Thus, only 15 patients (20.3%) of our cohort could have been included in MAIA trial considering the main exclusion criteria (cognitive disorders, ECOG PS > 3, Creatinine clearance < 30 mL/min).
The purpose of our study is not to assert that our therapeutic strategy is the reference ap-proach for super elderly/very frail patients. The PFS and OS are certainly insufficient, in particular for the 20% of patients for whom the results of the MAIA trial can be applied. However, these results constitute a basis for the evaluation of new therapeutic strategies, in particular those involving monoclonal antibodies (i.e. anti-CD38 monoclonal antibod-ies), and invite reflection on ways to improve management in a super elderly/very frail patient population, that is mostly excluded from prospective trials.
- In the introduction the Authors state that VMP and Rd are considered as the standard of care for newly diagnosed patients who are ineligible for ASCT but this statement is no longer true since DRd, D-VMP and VRd are the new standard therapies as suggested by the most recent ESMO guidelines (Dimopoulos et al, Ann Oncol 2021)
We thank reviewer 2 for this comment. We have updated the introduction to provide sufficient background and include all relevant references.
- It is true that management of octogenarian patients can be difficult due to comorbidities or altered performance status but we know that in frail patients, defined according to simplified frailty scale, who received DRd median PFS was NR vs 30.4 months in patients receiving Rd (Facon et al, Leukemia 2021, not mentioned in the paper). Safety profile was acceptable and infections leading to discontinuation were rare and pneumonia was a reason of discontinuation only in 2 patients in the frail group (1.2%) receiving DRd.
We have extensively modified the discussion to address this comment from reviewer 2. We are in full agreement with the data presented here. However, we have tried to emphasize that the population from MAIA trial is not the one we present. We have also added a paragraph on the limitations of using lenalidomide in super elderly/very frail patients. We have also detailed the limitations of the current definition of frailty according to the IMWG scores and the simplified frailty score.
- How many patients received Rd regimen in the same period and, in addition to peripheral neuropathy, what were other criteria for choosing Vd or Rd considering that this latter is an oral effective regimen, well tolerated and very easy to administer in an unfit population?
We cannot specify with certainty the number of patients who received revlimid, as the chemotherapy prescription software allowed a complete retrospective collection of patients treated with Bortezomib but not with velcade. However, we estimate that less than 5 newly diagnosed patients received this treatment. We have added a paragraph on the choices that motivated the use of bortezomib rather than lenalidomide : Our choice to propose a bortezomib dexamethasone doublet therapy over lenalidomide dexamethasone was based on several factors related to the specificities of our super elderly patients. An important issue is the adherence to oral cancer therapy in older adults. Adherence may be compromised specifically in the presence of cognitive impairment, multimorbidity, and polypharmacy. These relevant risk factors of poor adherence are particularly prevalent in our patients. Thus, the use of bortezomib instead of lenalidomide was a simple way to ensure adherence to treatments. In addition, nearly half of the patients had severe renal impairment. Although dose adjustment with lenalidomide were possible, we believed that the efficacy and toxicity profiles were uncertain in these patients, as was later demonstrated. Finally, even if the administration of bortezomib was more restrictive than lenalidomide, it indirectly induced a closer medical follow-up of these frail patients.
- I agree that all elderly patients should be treated (see the last statement of discussion section) but I think that these patients should receive the most effective regimen since if they relapse probably will not be able to receive further therapies
Reviewer 2 refers to the attrition rate by subsequent lines of therapy in patients with newly diagnosed multiple myeloma. Thus, in transplant ineligible NDMM patients, attrition is found to be as high as 50% per line of therapy. This would justify offering the most effective therapy upfront, as the most frail patients would not have the opportunity to receive additional therapies later.
We believe that this remains to be demonstrated, particularly in the most frail patients. Indeed, attrition data are primarily from US cohorts where treatment lines may be conditioned by insurance considerations. Secondly, when we look at the subgroup analyses of MAIA and ALCYONE, patients over 75 had an improved PFS with daratumumab, but there was no significant improvement in overall survival in the subgroup of patients over 75 years of age (which indirectly suggests that these patients were receiving effective subsequent lines of treatment that limited death by disease progression). Finally, the HOVON trial, which included only frail patients, showed disappointing survival results, mainly due to frequent treatment discontinuation in this frail patient population.

Round 2
Reviewer 2 Report
No comments